# Surgical Effect Observation and Treatment Strategy Analysis of Pseudo Urgency Syndrome

**DOI:** 10.3390/medicina58111506

**Published:** 2022-10-22

**Authors:** Zhenhua Gao, Han Lin, Kunbin Ke, Tingqiang Yao, Quan Zhang, Ling Li, Xingqi Wang, Jihong Shen

**Affiliations:** 1Department of Urology, The First Affiliated Hospital of Kunming Medical University, Kunming 650032, China; 2School of Mechanical and Electric Engineering, Kunming University of Science and Technology, Kunming 650093, China

**Keywords:** pseudo urgency syndrome, UUI, SUI, MUI, TOT, solifenacin succinate

## Abstract

*Background and Objectives*: pseudo urgency syndrome among patients with mixed incontinence (MUI) causes and the corresponding treatment strategies is explored. *Materials and Methods*: A total of 40 patients with MUI are treated with transobturator tape (TOT) and/or solifenacin succinate. Further, 30 patients with simple stress urinary incontinence (SUI) that were treated with transobturator tape (TOT) from the period of December 2018 to August 2020 are retrospectively analyzed; then, their clinical characteristics and therapeutic effects were summarized and analyzed. *Results*: The effective rates of SUI symptoms in MUI and simple SUI groups were 85% and 90%, respectively; further, the difference was noted as not statistically significant (*P* > 0.05). Among the 40 patients with MUI, 12 patients had unstable bladder contraction, and the other 28 patients showed normal bladder compliance. The treatment effectiveness rates of SUI symptoms in patients with unstable bladder contraction and normal bladder compliance were 83.3% and 85.7%, respectively; further, no significant difference was noted (*P* > 0.05). However, the effective rates of urge urinary incontinence (UUI) were 50% and 85.7%, respectively, however the difference was noted as statistically significant (*P* < 0.05). *Conclusions*: Most of the UUI symptoms in MUI patients may be “pseudo urgency syndrome” caused by the worry about the leakage of urine, rather than a real sense of UUI that is caused by excessive bladder excitement. Direct surgical treatment in patients with MUI can improve the symptoms of urinary incontinence, and the effect is more obvious in patients with urinary frequency who have normal bladder compliance according to urodynamics.

## 1. Introduction

Urinary incontinence is an involuntary leakage of urine that can cause symptoms in a wide range of severity and affect the patient’s quality of life. Symptoms could force major changes in the way the patient lives their life, including changes in the patient’s physical and mental health. The cost to treat urinary incontinence, either medically or surgically, is well over $10 billion per year [1].

The International Association of Urinary Control defined involuntary urinary leakage, in the presence of both urgent and increased abdominal pressure, as mixed urinary incontinence (MUI) [2].

Among women with urinary incontinence, approximately one-third have mixed incontinence, that is, an issue across all age groups with symptoms of both stress and urgency incontinence [3].

Stress incontinence is the involuntary loss of urine following increased intra-abdominal pressure or physical exertion (coughing, sneezing, jumping, lifting, exercising, etc.). Pseudo urgency syndrome is a specific type of MUI. It refers to a MUI with stress urinary incontinence (SUI) as the main pathological feature but presents as urge urinary incontinence (UUI) within a normal cystometrogram. Urge incontinence is the involuntary loss of urine preceded by a sudden and severe desire to pass urine. Bladder contractions may be stimulated by a change in body position (from supine to upright) or with sensory stimulation. The pathophysiology of urge incontinence is uninhibited bladder contractions caused by irritation or the loss in neurologic control of bladder contractions.

MUI may be urge predominant, stress predominant, or equal. The pathophysiology and treatment of MUI has lacked attention and in-depth research, especially in regard to treatment strategy. Urinary incontinence, particularly the mixed type, is an issue across all age groups. The purpose of this study, therefore, is to explore the causes of UUI symptoms in MUI patients and to analyze the treatment methods of MUI.

## 2. Materials and Methods

### 2.1. Research Object and Preoperative Preparation

#### 2.1.1. General Information

This study selected 70 patients with urinary incontinence who were admitted to the Department of Urology of the First Affiliated Hospital of Kunming Medical University from December 2018 to August 2020. This included 30 patients with simple SUI and 40 patients with MUI. The inclusion criteria were as follows: (1) The clinical diagnosis was either SUI or MUI, and (2) all patients agreed to the operation plan and completed the questionnaire by themselves. The exclusion criteria were as follows: (1) There are serious contraindications to surgery, such as severe coagulation, dysfunction, or functional changes in important organs, (2) urodynamic examination suggested neurogenic bladder or bladder detrusor weakness, and (3) patients with urinary tract infection, unknown consent, and/or failure to understand the content in the follow-up. There was no significant difference in age, height, weight, marriage, and childbearing history. There was also no significant difference in the follow-up time between the two groups (*P* > 0.05, Table 1) either. All patients filled out the International Urinary Incontinence Advisory Committee urinary incontinence questionnaire short form (ICI-Q-SF) and also the overactive bladder scale score (OABSS) form.

#### 2.1.2. Urodynamic Examination

Urodynamic examination was performed in both groups before operation.

#### 2.1.3. Ultrasonography

All patients underwent ultrasonography before operation. The residual urine volume was less than 15 mL and there were no cases of hydronephrosis.

### 2.2. Treatment and Evaluation

#### 2.2.1. Therapeutic Method

In this study, the two groups of patients underwent transobturator tape (TOT) treatment. A disposable, middle urethral suspension sling produced by Herniamesh (Herniamesh S.r.l., San Mauro, Italy) was used and the same doctor performed the TOT operation under general anesthesia [4]. Surgical procedure: The patient took the lithotomy position, indwelling the urinary tube, and sequentially had 40 mL of 0.9% normal saline injected into the anterior wall of the vagina in order to form a water pad. A 3 cm long median straight incision was made downward from 1 cm below the external orifice of the urethra. The space between the vagina and urethra was separated from the dorsal side of the pubic descending branch to the posterior side of the pubic bone. A horizontal line was made at the base of the flat clitoris, 0.5 cm beyond the border of the lateral border of the descending pubic ramus. The spiral puncture needle was inserted from the outside, close to the descending branch of pubic bone, and passed through the obturator membrane. The index finger guided the puncture needle to pass through the gap between the vagina and urethra and took out one end of the sling, and also in the process checking that the urine color was clear and the mucosa of the vaginal side wall was complete. Next, the other side was punctured in the same way to draw out the other end of the sling; then, the protective cover was removed and the tension of the sling was also adjusted. The selvage suture incision was then closed, and the vaginal iodophor gauze was also retained. The next day, the vaginal iodophor gauze and urinary tube was pulled out, and the patient was instructed to urinate on their own.

After operation, the patients were instructed to perform pelvic floor function training. For patients with UUI symptoms, who still had clinical symptoms after operation, solifenacin succinate was given once a day, 5 mg each time, orally for two weeks to one month. For patients with severe symptoms [5], solifenacin succinate was given twice a day.

#### 2.2.2. Postoperative Follow-Up and Evaluation

All patients included in the study were followed up with in a period lasting from 2 to 19 months. The patients were followed up with by telephone to ask if there were any complications, such as infection at the operation site, difficulty in urination, pain at both sides of the thigh, exposure of the sling and erosion, etc. The ICI-Q-SF scale and OABSS questionnaire were filled out by the patients and this was then compared with the preoperative condition of the patients. The ICI-Q-SF score, including number of urine leakage (FOL), volume of urine leakage (VOL), and quality of life impact (QOL).

The evaluation criteria used to define the improvement of postoperative urinary incontinence symptoms were based on the ICI-Q-SF scoring table. These criteria were noted as: (1) Complete remission—postoperative VOL = 0 and QOL ≤ 3 for SUI symptoms; (2) partial remission—postoperative VOL = 2 and QOL ≤ 3 for SUI symptoms; and (3) failure—postoperative VOL > 2 and/or QOL > 3 for SUI symptoms. The evaluation criteria for urgent symptom improvement were compared with the preoperative and postoperative OABSS questionnaire answers. These included: (1) complete remission—the postoperative OABSS questionnaire indicated that there were no symptoms of urgency and UUI; (2) partial remission—the postoperative OABSS questionnaire showed that the severity of OAB symptoms was significantly improved (severe OAB improved to mild OAB); (3) The rest were failures.

### 2.3. Statistical Analysis

SPSS 22.0 statistical software (IBM Corp., Armonk, NY, USA) was then used to process the data. All measurement data were expressed by the means ± standard deviation (x ± s), and the *t*-test was used for the purposes of comparison between the two groups. The counting data were expressed by the rate, and the chi-square test was used for the purposes of comparison between the two groups. The difference was understood as statistically significant if *P* < 0.05.

## 3. Results

### 3.1. Analysis of Postoperative Curative Effect

After postoperative follow-up, the effective rates of symptom treatment in the two groups are as follows: (1) In 30 patients with simple SUI, 22 patients (73.3%) achieved complete remission and 5 patients (16.7%) achieved partial remission after receiving TOT treatment and pelvic floor functional training. The effective rate was 90%; (2) A total of 40 patients with clinically diagnosed MUI received TOT treatment and pelvic floor function training, and those with persistent symptoms—such as frequent and urgent urination after operation—were given solifenacin succinate once a day, 5 mg each time, orally for two weeks to one month. After one month of treatment, 24 cases (60%) had complete remission of SUI symptoms, 10 cases (25%) had partial remission, and the effective rate was 85%. A total of 16 cases (40%) had complete remission of UUI symptoms, 11 cases (27.5%) had partial remission, and the effective rate was 67.5%. The effective rate of SUI symptom treatment in the two groups was statistically compared (*P* > 0.05), and the difference was noted as not statistically significant (Table 2).

By analyzing the urodynamic examination results of 40 patients with MUI, we found that 12 patients had mixed incontinence signs, such as unstable bladder contraction, which is to say that the patients were urodynamically positive and accounted for 30% (12/40). This then means that the remaining 28 patients had no mixed incontinence signs following urodynamic examination, which is to say they were urodynamically negative, and subsequently accounted for 70% (28/40). After treatment, the effective SUI symptom rates of the two groups were 83.3% and 85.7%, respectively. Thus, the curative effects were obvious to see. Further, there was no statistical difference between the two groups (Table 3). In addition, the effective rates of UUI symptoms were 50.0% and 85.7%, respectively. Further, statistical significance was noted (Table 4).

### 3.2. Comparison of the Quality-of-Life Scores of Postoperative Patients

The ICI-Q-SF symptom score, the ICI-Q-SF QOL score of 30 patients with simple SUI, and the 40 patients with symptomatic MUI after operation were all compared, respectively. The results showed that there was no statistical difference between the two groups of patients after operation. (*t*-test, *P* > 0.05; see Table 5).

### 3.3. Comparison of Postoperative Complications

All patients were followed up with postoperatively. Further, there were 15 cases of complications, which are noted as follows: Pain and foreign body discomfort were found in 9 cases, which were inclusive with 2 cases of SUI and 7 cases of MUI. However, all the symptoms disappeared within half a month after operation; There were 5 patients with mild dysuria after operation, which were inclusive with 2 patients with simple SUI and 3 patients with MUI, however there were no symptoms of acute urinary retention; In addition, one patient with MUI had incision bleeding after operation; Lastly, the other patients had no bladder perforation during operation and no complications such as urinary retention and sling exposure after operation. The Clavien–Dindo classification of postoperative complications in the two groups is shown in Table 6.

## 4. Discussion

Urinary incontinence is a kind of disease that can seriously trouble the daily life of many women. The most common forms of urinary incontinence are SUI and UUI. Among them, SUI patients often have UUI symptoms, which is to say MUI. Compared with simple SUI, the mixed urgent symptoms of MUI have a more serious impact on the psychological distress and daily life of patients [6]. However, there are still many disputes about the etiology and treatment strategy of MUI.

The prevalence of MUI ranges from 0.6% to 59.2% [7], while in SUI cases, the incidence rate of UUI symptoms can be as high as 69.3% [8]. In this group of cases, 40 patients were clinically diagnosed with MUI, accounting for 57.1% of all patients with SUI symptoms. If MUI was diagnosed only by urodynamic bladder compliance, low volume, and uninhibited contraction then the proportion was only 17.1%. It can now be seen that there is a great difference between the subjective symptom diagnosis of patients and the prevalence of MUI as diagnosed by urodynamics. Some studies believe that this difference is caused by different research objects, or differences in patient statements, diagnostic standards, or even other factors [9]. However, the high prevalence of UUI caused by this difference is obviously unconvincing.

In this study, after urodynamic examination of 40 enrolled MUI patients, only 12 patients had MUI urodynamic changes such as unstable bladder contraction. The urodynamics of the remaining 28 patients showed negative signs of MUI, which indicated that most of the MUI patients diagnosed by symptoms had normal bladder compliance, suggesting that they may have urgent symptoms caused by mental factors of the patients that could be caused by SUI. Digesu et al. [10] performed urodynamic analysis on 1626 patients with UUI and SUI. These patients had symptoms such as frequent urination and urgency of urination; it was found, however, that only 18% of the patients’ urodynamic test results suggested the presence of UUI, which could be diagnosed as MUI. Lin et al. [11] performed urodynamic examination on 340 patients with lower urinary tract symptoms and found that only one patient could be diagnosed as possessing MUI. These results are consistent with our study. The reason for these type of results may be related to the lower degree of UUI in MUI patients, or in the symptoms of urgency of urination that originate from the urethra, which cannot be detected by urodynamic examination [12]. However, this latter inference does not seem to fully explain this phenomenon.

In this study, we found that the patients whose clinical symptoms were MUI but whose urodynamic examination was otherwise normal—all had psychological and behavioral UUI symptoms of the fear of urination overflow. That is to say, in order to avoid leakage of urine, SUI patients urinate immediately when they have the intention to urinate for the first time. Thus, this formed a habit of repeated urination, resulting in enhanced self-urination awareness, therefore forming a “pseudo” UUI, following their normal urodynamic examination. In addition, the patient also retrained their bladder due to the fear of repeated urination from urinary incontinence, which further aggravated the symptoms of frequent urination and urgency. Osman et al. [13] named this symptom of frequent urination and the subsequent urgency to avoid inducing SUI as “pseudo urgency syndrome”. To sum up, we can conclude that the actual proportion of MUI patients who, in reality, have the pathological basis of UUI is low, because of the interference of patients with “pseudo urgency syndrome”. There are great differences in the epidemiological investigation of MUI in different regions, resulting in a higher proportion of MUI patients in SUI. The real cause of MUI is mental UUI caused by SUI.

As the mechanism of simple SUI is mainly stress bladder dysfunction, the most common causes are vaginal delivery and age, which lead to a weak pelvic floor support structure, lax bladder neck closure when abdominal pressure increases, and shorter functional urethra—thus SUI occurs. The pathological basis of UUI is high bladder excitability or detrusor overactivity and poor compliance [14]. Therefore, it is difficult to explain these with the same pathogenic factor or mechanism, which will inevitably lead to disputes over treatment methods.

Restoring the original anatomical structure of the pelvic floor through surgery is the main method to treat SUI at present, and its short-term cure rate can reach more than 90% [15]. However, it is still controversial whether MUI patients should undergo surgical treatment first. The 2017 edition of urinary incontinence guidelines in China [16] advised that patients with MUI should be treated with caution. In particular, those with MUI who are dominated by UUI should be treated with drugs in a conservative manner first, and those who fail to respond to conservative treatment should then be treated with surgery. This is because it has been reported that 40% of patients with MUI will continue to have frequent urination and urgent urination after surgery [17]. However, recent studies have found that the urinary incontinence symptoms of MUI patients can be greatly improved after surgical treatment, and there is no obvious tendency of deterioration. Natale et al. [18] treated 86 patients with urinary incontinence with TOT. After an average follow-up of 59 months, they found that the cure rates of SUI and MUI patients could reach 83.7% and 74.4%, respectively. Padmanabhan et al. [19] treated 487 MUI patients with TOT, and the subjective improvement rate of patients reached 72.8%. A medium and long-term study by Zhang et al. [12] found that the cure rate of UUI could also reach 76.92% if only TOT was given to MUI patients, and the curative effect remained stable for a long time. The quality-of-life of patients was significantly improved. A 5-year follow-up study by Yonguc et al. [20] also showed that the effective rate of TOT in treating MUI patients was 83.3%.

In this study, the effective rate of SUI symptom treatment in MUI patients after surgical treatment was similar to that in simple SUI patients, i.e., as high as 85%. It should be noted that 28 patients with urodynamic “Negative” signs of MUI showed significant improvement in UUI symptoms after treatment and did not require long-term drug intervention. Therefore, for patients with “pseudo urgency syndrome”, the real cause is SUI. When SUI is cured, UUI symptoms will naturally be alleviated. Patients with non-inhibitory contractions in other urodynamic examinations can also obtain long-term satisfactory effects through pelvic floor rehabilitation training and drug treatment. Therefore, for MUI patients, surgery should be performed first in order to solve SUI symptoms, so that patients can eliminate the psychological worries of fear of urine leakage. Thus, patients dare to hold their urine, reducing the training stimulation of repeated urination, and improving or even eliminating UUI symptoms.

## 5. Conclusions

In conclusion, the prevalence of the clinical diagnosis of MUI is increasing due to the interference of “pseudo urgency syndrome” and other factors. Most UUI symptoms in MUI are caused by mental factors, and urodynamic examination is an important means for differential examination. For MUI patients, surgery should be undertaken first in order to treat SUI. Pelvic floor rehabilitation training and drug therapy are then feasible, if there are still urgent symptoms.

## Figures and Tables

**Table 1 medicina-58-01506-t001:** Comparison of general data between simple SUI and MUI patients.

Characteristics	Simple SUI (*n* = 30)	MUI (*n* = 40)	*P* Value
Age (mean ± SD, in y)	52.57 ± 7.67	51.05 ± 8.72	0.451
BMI (mean ± SD, in kg/m^2^)	23.14 ± 2.23	24.08 ± 3.31	0.081
Number of pregnancies (mean ± SD, in no. of times)	3.17 ± 1.98	3.45 ± 1.99	0.557
Parity (mean ± SD, in no. of times)	1.73 ± 1.14	2.08 ± 1.16	0.225
Follow up (mean ± SD, in months)	1.73 ± 1.14	8.8 ± 5.59	1.000

Abbreviations—SD: standard deviation; BMI: body mass index; SUI: stress urinary incontinence; MUI: mixed urinary incontinence; and y: year.

**Table 2 medicina-58-01506-t002:** Effective rate of postoperative SUI symptom treatment in simple SUI and MUI patients. (Case expressed in (%)).

Group	Effective (Case)	Invalid (Case)	Total (Case)	Effective Rate (%)
Simple SUI	27	3	30	90
MUI	34	6	40	85

(*X*^2^ = 0.066 and *P* = 0.797. The difference was not statistically significant.).

**Table 3 medicina-58-01506-t003:** Effective rate of postoperative SUI symptoms in patients with positive urodynamics and patients with negative urodynamics. (Case expressed in (%)).

Type	Effective (Case)	Invalid (Case)	Total (Case)	Effective Rate (%)
Urodynamic positive	10	2	12	83.3
Urodynamic negative	24	4	28	85.7

(*X*^2^ = 0.000 and *P* = 1.000. The difference was not noted as statistically significant.).

**Table 4 medicina-58-01506-t004:** Effective rate of postoperative UUI symptoms in urodynamic positive patients and urodynamic negative patients. (Case expressed in (%)).

Type	Effective (Case)	Invalid (Case)	Total (Case)	Effective Rate (%)
Urodynamic positive	6	6	12	50.0
Urodynamic negative	24	4	28	85.7

(*X*^2^ = 3.968 and *P* = 0.046. The difference was noted as statistically significant.).

**Table 5 medicina-58-01506-t005:** Comparison of postoperative quality of life scores between simple SUI and MUI patients. (x ± s).

Group	ICI-Q-SF Symptom Score	ICI-Q-SF QOL Score
Simple SUI (mean ± SD)	1.71 ± 2.85	1.32 ± 2.51
MUI (mean ± SD)	2.15 ± 3.17	1.53 ± 2.61
*P* value	0.547	0.742

Abbreviations—SUI: stress urinary incontinence; MUI: mixed urinary incontinence; ICI-Q-SF: the International Urinary Incontinence Advisory Committee urinary incontinence questionnaire short form; and QOL: quality of life impact.

**Table 6 medicina-58-01506-t006:** Clavien–Dindo classification of postoperative complications in patients with simple SUI and MUI.

Group	Case	Clavien–Dindo Classification				
		Stage I	Stage II	Stage III	Stage IV	Stage V
Simple SUI	4	4	0	0	0	0
MUI	11	10	0	1	0	0

## Data Availability

Not applicable.

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
