# Peer review of "Surgical Effect Observation and Treatment Strategy Analysis of Pseudo Urgency Syndrome"

_medicina, 2022, doi:10.3390/medicina58111506_

Round 1

Reviewer 1 Report

The article brings into discussion a problem that practitioners frequently have to face and treat, and as the authors say affects women of middle age and for which treatment may include surgical and medical methods. The authors want to highlight the "pseudo-emergency syndrome" but in introduction it would be useful to define what pseudo-emergency syndrome means and what are its characteristics so readers understand the difference between SUI, MUI, UUI and pseudo-emergency syndrome. 

In treatment evaluation section, in line 102-104 the authors describe that patients still having symptoms after surgery receive oral treatment once a day or twice a day from two to four weeks taking into account the severity of symptoms.  How the severity of the symptoms is quantified should be explained, so it would be easier for the readers. Also from line 111-120 the authors use shortenings like VOL =0 or VOL above 2 without explaining what "VOL" means. 

In statistical analysis section, line 126, for statistically significance >0.05. In all the rest of the text p>0.05 is considered without statistical significance. Maybe it is just a writing mistake, but it should be corrected. 

 In results sections, table 4 should be more detailed explained, for invalid and valid subjects. Are these the patients who recieve oral tratment?

As strong points for this article: first of all, the subject and defining the pseudo-emergency syndrome, the title which highlights the study, another strong point can be considered the number of patients included in this study. The references of this study are mostly recent publications, and relevant for this article. In the section discussions the articles that sustain the results of this study are relevant and including a large cohort of patients. 

Reviewer 2 Report

 The paper is interesting and in accord with the literature is quite complete.

Contribution of all authors is significant and could be interesting for scientist.

Analysis and data interpretation are adequacy, writing style could be improving.

The only suggestion is you could add a drawing this it would make reading the manuscript more appealing.

Moreover, I suggest improving how a tailed and multidisciplinary approach is needed by citing:

 -PMID: 32485429

-PMID: 32446632

-PMID: 33397168
